# Exosome as a Delivery Vehicle for Cancer Therapy

**DOI:** 10.3390/cells11030316

**Published:** 2022-01-18

**Authors:** Bohyun Moon, Suhwan Chang

**Affiliations:** 1Asan Medical Center, Department of Internal Medicine, University of Ulsan College of Medicine, Seoul 05505, Korea; hyunbo35@gmail.com; 2Asan Medical Center, Department of Biomedical Sciences, University of Ulsan College of Medicine, Seoul 05505, Korea; 3Asan Medical Center, Department of Physiology, University of Ulsan College of Medicine, Seoul 05505, Korea

**Keywords:** exosome, cell-derived vesicles, drug delivery, cancer therapy

## Abstract

Exosomes are small extracellular vesicles that are naturally produced and carry biomolecules such as proteins, microRNAs, and metabolites. Because of their small size and low level of biomolecule expression, the biological function of exosomes has only been identified recently. Despite the short history of investigation, exosomes seem to have remarkable potential as a delivery vehicle. With regards to cancer therapy, numerous antitumor agents demonstrate serious side effects (or toxicity), which has led to the unmet need for improving their selectivity and stability. Exosomes, either produced naturally or generated artificially, provide an attractive platform to load many types of molecules such as small molecules, biologics, and other therapeutic agents. Furthermore, the features of exosomes can be designed by selecting their source cells, or they can be engineered to incorporate affinity tags; thus, exosomes show promise as effective delivery vehicles for the complex tumor microenvironment. In this review, we focus on various exosomes produced from different cell types and their potential uses. Moreover, we summarize the current state of artificial exosomes as a drug carrier and provide an overview of the techniques used for their production.

## 1. Introduction

### 1.1. Drug Delivery Vehicles for Cancer Therapy

Cancer is the second leading cause of death globally, with a high mortality rate, causing 9 million deaths annually, and approximately 18.1 million new cases are identified every year [1]. Current cancer treatment options include surgical intervention, chemotherapy, and radiation therapy or a combination of these options [2]. Chemotherapy is one of the most widely employed clinical cancer treatments, which works by interfering with DNA synthesis and mitosis, leading to the death of rapidly growing and dividing cancer cells. These agents are nonselective and can damage normal tissues, causing severe undesired side effects such as nausea and vomiting. In fact, the severe adverse effects induced by chemotherapeutic drugs on normal tissues and organs are a major reason underlying the high mortality rate of patients with cancer [3]. Additionally, because of the poor tissue penetration of these drugs, higher doses are required, leading to elevated toxicity in normal cells. Therefore, it is desirable to develop chemotherapeutics that can effectively reach the target cancerous cells, thereby reducing adverse effects while improving therapeutic efficacy.

In the last few years, numerous attempts have been made to develop drug delivery systems (DDSs) with improved therapeutic efficacies. The use of nanotechnology has had a profound impact on clinical therapeutics. Compared with conventional chemotherapeutic agents, nanoscale drug carriers have several advantages; that is, they improve treatment efficacy while avoiding toxicity in normal cells due to features such as highly selective accumulation in tumors via the enhanced permeability and retention effect and active cellular uptake [4,5]. An active targeting approach can be achieved by binding nanocarriers, including chemotherapeutic agents, to molecules that bind to overexpressed antigens [6]. Most drug delivery vehicles are chemically synthesized using lipids or lipid-like molecules. Despite the remarkable advances and successes in the design and effectiveness of synthetic drug vehicles, some limitations to their practical application exist. The main disadvantages are their toxicity and low biocompatibility [7]. To overcome these limitations, there is increasing recognition of natural drug delivery vehicles due to their advantages of evasion of the host immune system and high efficacy of entering target cells. In the past, bacteria, viruses, red blood cells, and lymphocytes have been considered possible natural drug delivery vehicle candidates [8]. Recently, exosomes have attracted attention as novel DDSs. There has been a growing interest in exosome research in the last decade due to their emerging role as intercellular messengers and their potential in managing disease [9].

### 1.2. Basic Properties of Exosomes

Exosomes are a type of cell-derived vesicles characterized as extracellular vesicles (EVs). EVs are nanometer-sized small membrane vesicles secreted by most cells, containing proteins, lipids, and nucleic acids, which are specific to their cell origin [10]. EVs are categorized into three types—exosomes, apoptotic bodies, and microvesicles [11]. The difference between these EVs is thought to be due to biogenesis, which in turn determines the cargo contents and functions. Microvesicles are formed from the budding of the cell membrane, whereas exosomes are the result of endocytosis from multivesicular bodies (MVBs) that eventually fuse with the plasma membrane and are then released to the extracellular space [12]. Exosomes, with a diameter in the range of 40–100 nm, possess a lipid bilayer membrane with the same orientation as the plasma membrane and carry cargo that includes both proteins and genetic material [13]. Exosomes have an array of constituents such as surface proteins, heat shock proteins (HSPs), lysosomal proteins, tumor-derived genes, fusion proteins, and nucleic acids, each exhibiting certain functions. The lipid bilayer of the exosome is rich in cholesterol and diacylglycerol [14]. Lipids such as sphingomyelin and monosialotetrahexosylganglioside determine the rigidity of the exosomes. In addition, different types of phospholipid transportation enzymes in exosomes are expressed by phosphatidylserine [15]. Exosomes have similar components due to their endosomal origin, including HSPs, membrane transporters (annexins, Rab GTPases, and flotillin), and MVB proteins, including TSG101, Alix, integrins, and tetraspanins (CD9, CD63, CD81, and CD82), which mediate signaling, cell fusion, and migration [16].

Exosomes contain various nucleic acids. Messenger RNA (mRNA) is the mediator of horizontal transfer of genetic information in exosomes [17], whereas micro RNA (miRNA) serves the function of cell targeting and gene silencing [18]. Exosomes also have noncoding RNA, the shorter ones of which regulate gene expression [19], and long noncoding RNAs are involved in carcinogenesis and cancer progression [20]. Circulating DNA (cDNA), a heterogenous population of genomic and mitochondrial DNA, contains genetic alterations and reflects mutations, rearrangements, and amplifications in tumor tissues [21]. Exosomes have been reported to be involved in several processes such as cell–cell communication through the exchange of proteins and genetic materials, immunomodulatory functions, antigen presentation, tumor growth suppression, endothelial cell migration, and inflammation. The main function of exosomes is intercellular communication by transferring lipids, RNA, and cytosolic proteins. This finding indicates the possibility of using exosomes as DDSs to deliver therapeutic drugs.

Exosomes are produced by most cell types, including dendritic cells (DCs), neutrophils, epithelial cells, and tumor cells. They are also found in biological fluids [22]. Depending on the cell of origin, exosomes contain cell-specific proteins and lipid constituents that reflect their cellular source origin [23]. Furthermore, because of their stability, exosomes are widely distributed in biological fluids such as the blood, urine, bronchoalveolar lavage fluid, breast milk, amniotic fluid, synovial fluid, and ascites [23]. These properties suggest that exosomes are attractive vehicles for drug delivery. Mesenchymal stem/stromal cells (MSC)-derived exosomes do not contain class I and class II human major histocompatibility complex (MHC) proteins or co-stimulatory molecules such as CD80 and CD86, which helps them evade the human immune system. From the immunological perspective, MSC-derived exosomes are mostly used nowadays [24].

## 2. Natural Cell-Type Specific Exosomes

Numerous different cell types such as DCs, mast cells (MCs), B cells, T cells, platelets, and tumor cells are known to secrete exosomes (Figure 1) [25]. Exosomes released from tumors have been widely studied in various cancer types, such as renal cancer, breast cancer, and melanoma. Tumor cells continuously secrete membrane vesicles into the extracellular environment. Exosomes released by malignant tumor cells contain specific proteins, lipids, DNA molecules, miRNAs, mRNAs, and noncoding RNAs, which are important for cancer cell communication with the environment [26]. Tumor-derived exosomes or tumor-related exosomes are considered to be closely associated with the pathogenesis and microenvironmental formation of cancer because the number of exosomes in cancer cells is higher than that in normal cells [27].

By contrast, DCs play a central role in initiating antigen-specific immunity and tolerance [28]. In cancer, DCs act as the initial link between oncogenesis and the host immune system, which is the first step of the immunity cycle that aims to eliminate cancer cells through the activation of T cells. DC-derived exosomes are nanometer-sized membrane vesicles that are secreted by the antigen-presenting cells of the immune system. DCs secrete a large number of exosomes to induce effective anti-cancer effects. DC-derived exosomes containing MHC I, MHC II, CD86, and HSP70/HSP90 chaperones can trigger CD4+ and CD8+ T cell activation. Under costimulation of secreted IL-2 and exosomal peptides, MHC I is passed to CD8+ T cells and induces more effective antitumor immunity in vivo [29,30,31].

As the source of immune cell-derived exosomes, NK cells contribute to immunosurveillance and function as the body’s first line of defense against several human disorders, including pathogen infections and cancers. NK cells can directly recognize and effectively kill oncogenic transformed cells that are normally devoid of class I MHC antigen expression, thus participating in anti-cancer immunity [32]. NK cell-derived exosomes also harbor prototype NK markers and killer proteins [33]. Additionally, NK exosomes can exert their cytolytic activity by directly diffusing into tumor tissues and subsequently overcoming the homing deficiency of NK cells to tumor sites [34]. In addition to exosome-specific markers (e.g., tsg 101, CD81, CD63, and CD9), NK cell markers (NKG2D, CD94, perforin, granzymes, and CD40L) are also expressed in NK-derived exosomes, which are both involved in cytotoxicity and immune responses. These exosomes can induce target cell death by multiple killing mechanisms [35,36].

MC is an important component of the innate immune system and plays a crucial role in Th2 responses [37]. MCs can secrete exosomes that display biological functions in RNA and protein transfer, intercellular communication, and immune regulation [38]. MC-derived exosomes can affect the biological functions of DCs, T cells, and B cells [38,39]. For example, CD63+ and OX40L+ exosomes from MCs promote the proliferation and differentiation of CD4+ Th2 cells via the OX40L–OX40 interaction [40]. MC-derived exosomes also induce immature DCs to upregulate MHC II, CD40, CD80, and CD86 expression and to confer the antigen-presenting capacity to T cells, thereby leading to the initiation of antigen-specific immune responses [41]. Similarly, neutrophil-derived exosomes also contain proteins, mRNA, and miRNAs, which are associated with inflammatory reactions, immune response, and cell communication [42,43,44]. They can affect the activity of other immune cells, such as macrophages, by transferring several proinflammatory factors [45]. These exosomes have been reported to bind and degrade extracellular matrix (ECM) via integrin Mac-1 and neutrophil elastase, consequently leading to inflammatory disease progression [46].

As another exosome source, MSCs are multipotent nonhematopoietic adult cells, discovered by Alexander Friedenstein [47]. MSCs, possibly originating from the mesoderm, were reported to express CD73, CD90, and CD105 plasma membrane markers, and not CD14, CD34, and CD45 [48]. Relative to other cell types, MSCs possess distinct advantages as an exosome source. They release higher numbers of exosomes than other cells. MSC-derived EVs are relatively well tolerated in different animal models and show more stability and sustainability in human plasma [49].

## 3. Artificial Exosomes as a Drug Delivery Vehicle

Exosomes have been suggested to be ideal DDSs with potential for application in a broad range of pathologies, including cancer, because of their organotrophic properties [26]. However, the low yield, high cost, and laborious methods of production of cell-derived exosomes are limitations, together with the lack of standardization for relevant processes [50]. Recently, artificial exosomes have been developed to overcome the drawbacks of natural exosomes as new theragnostic biomaterials for potential clinical applications [51]. A recent study reported the incorporation of CRISPR/gRNA into exosome [52]. In addition, siRNA, aptamer, and antisense oligonucleotide can be delivered via exosomes [53]. Despite promising results of exosome-mediated drug delivery, the translation of exosomes is challenged by massive production, purification, modification, drug loading, and storage. Because of the shortcomings of natural exosomes, a growing number of studies are aiming to develop artificial exosomes using the top-down, bottom-up, or biohybrid approach. The development of artificial exosomes, which have the advantages of both natural and synthetic nanoparticles, through nanobiotechnology holds great promise for advanced drug delivery.

### 3.1. Limitations of Artificial Lipid Bilayer Nanoparticles

When drug-loaded synthetic nanoparticles enter the bloodstream, there are two main issues with drug nanoformulations: toxicity and rapid clearance by the mononuclear phagocyte systems. Macrophages in the reticuloendothelial system (RES), located in the liver and spleen, take up particles bound with serum proteins [54]. Several efforts have been made to overcome this clearance of particles and improve distribution in vivo. The most widely used method is the steric stabilization of the liposomal surface by using polyethyleneglycol (PEG) [55]. It is hypothesized that PEG on the surface of liposomes attracts a water shell, resulting in reduced adsorption of opsonins and recognition of the liposomes by the mononuclear phagocytic systems [56]. This, in turn, leads to extended circulation time and improvement in tumor delivery. However, although PEGylation decreases clearance by the MPS, it reduces the interaction of the nanoformulation with target and barrier cells, thus decreasing the drug biodistribution in diseased tissues. Furthermore, PEG induces antibody-related immune reactions and accelerates blood clearance [57,58,59]. Moreover, surface modification of nanoparticles using CD47 or peptide derivatives from this marker, termed the “don’t eat me” signal, has proven effective for enhancing drug delivery [60].

### 3.2. Advantages of Artificial Exosomes Compared to Artificial Lipid Bilayer Nanoparticles

Compared to artificial, human-engineered nanoparticles, as natural nanovesicles, exosomes are good candidates for drug delivery due to their low immunogenicity and ability to enter tissues. Exosomes have advantages of both synthetic nanocarriers and cell-mediated drug delivery, avoiding the rapid clearance and toxicity associated with synthetic vehicles, as well as the complexity in utilizing cell-mediated DDSs in the clinic. These unique features make exosomes an attractive option for use as a drug delivery vehicle for cancer treatment. While artificial nanoparticles cannot pass the blood–brain barrier, endothelium, cell, and tissue barriers, exosomes have the natural ability to cross the normal blood–brain vascular barrier by transcytosis [61,62]. Thus, they are available for systemic treatment of CNS-inflammatory disorders and possibly cancers. Furthermore, exosomes have great resistance to various noxious environments. Exosomes resist the stomach acid and can likely also survive in phagolysosomes after cellular uptake and can resist the harsh tissue conditions of hypoxia [63,64]. These characteristics enable exosomes to function in the combined acidic and hypoxic environments of cancers and other types of tissue necrosis. Exosomes can naturally and easily evade the RES and avoid immune detection. Thus, they have a long in vivo duration of action. Furthermore, artificial nanoparticles demonstrate poor penetration of solid tumors and tissue-inflammatory infiltrates. However, exosomes can naturally penetrate tissues that have dense inflammation to target particular cells without any alterations for subsequent specific affinity targeting of target cells [62].

### 3.3. Challenges Associated with Artificial Exosomes Compared to Lipid Bilayer Nanoparticles

Despite the several advantages of exosomes as drug delivery vehicles, the application of artificial exosomes is still challenging in terms of massive production, standard purification protocols, cargo loading, storage stability, and modification cost. Because physical and biological stability is typically limited to a shorter time period, the International Society of EVs recommends storage at −80 °C in phosphate-buffered saline [65]. However, this storage condition is unfavorable in terms of energy consumption, transportation, and, most importantly, clinical application. Generally, freezing–thawing is considered to destabilize EVs, for example, by changing the EV morphology, function, particle size, and concentration [66]. Freezing–thawing studies have revealed improved colloidal EV stability in the presence of sucrose or potassium phosphate buffer instead of sodium phosphate buffer or phosphate-buffered saline [67]. Less aggregation and/or vesicle fusion occur at neutral pH than at slightly acidic or alkaline pH. In addition, the purification method is time-consuming. Some EVs are similar to exosomes in their physical properties, such as size and density, which makes the isolation of exosomes considerably challenging. Therefore, it is hard to produce and purify exosomes on a large scale [51], making it one of the active areas of research as described below.

## 4. Purification and Drug Loading of Exosomes

### 4.1. Approaches for the Isolation of Exosomes

To use exosomes as biomarkers and DDSs, their isolation, purification, and characterization are important and can be improved by innovative technologies. Numerous methods have been developed to facilitate the isolation of exosomes from biological resources. Ultracentrifugation is the gold standard of exosome isolation (Figure 2). Ultracentrifugation is based on the sedimentation coefficient difference between exosomes and other extracellular content. Under certain centrifugal forces, different extracellular components of fluidic samples can be sequentially separated based on the density, size, and shape. Among them, recently, density gradient ultracentrifugation has achieved the purest exosome samples. However, this method is time-consuming since it takes a while to attain the equilibrium of solutions [68,69]. Ultrafiltration is a membrane separation technique based on the size and molecular weight of exosomes and other contents. Exosomes can be separated from macromolecules using membranes containing pores equivalent to exosomes with a size of 100 nm so that they pass through, and other contents are retained on the membrane. Multiple steps of membrane washing increase the processing time. However, compared with the ultrafiltration method, ultrafiltration-based exosome isolation dramatically shortens the processing time and does not require special equipment, presenting an ideal substitute to the classical ultracentrifugation strategy [70]. The principle of immunological separation is based on the antigen–antibody reaction to capture exosomes (Figure 2). This method exploits the presence of various proteins on exosome membranes to capture them. Recent studies have focused on antibody-coated plates, chromatography matrices, and beads for immunological separation with high purity and less time consumption. It is an expensive method, as it involves special reagents and cell-free samples and limits the use of large-scale samples [68].

Among various isolation methods for EV, size exclusion chromatography (SEC) is considered an effective way to obtain homogeneous EVs [71]. SEC is also reported to remove soluble protein contaminants and is relatively easy to scale up for manufacturing clinical-grade products [72]. For clinical trials of exosomes, a frequently applied method is tangential flow fractionation combined with ultracentrifugation, as indicated in a recent report [73]. This method can maximize the purity, uniformity, and integrity of the exosomes.

Table 1 summarizes the pros and cons of the routinely used three exosome isolation methods [74].

### 4.2. Approaches for Drug Loading on Exosomes

Methods for encapsulating cargo into exosomes can be divided into two types: cell-based loading methods and non-cell-based loading methods. In the cell-based loading approach, cargo is usually delivered into the donor cells first. After being packaged into EVs, the cargo can be secreted and collected in an EV-carrying manner for therapeutic use [75]. The non-cell-based loading approach involves directly loading drugs into the isolated EVs through electroporation, sonication, incubation, and/or transfection [76]. Table 2 summarizes various exosome drug loading methods. Considering previous results of measured efficiency, sonication seems to work well in macrophage-derived exosomes, whereas electroporation seems better for primary DC-derived exosomes [77].

## 5. Therapeutic Aspects of Exosomes as a DDS

### 5.1. Exosomes: The Natural Drug Delivery Vehicle

Exosomes have benefits as drug delivery vehicles, such as tissue specificity, safety, and stability. They can deliver their cargo across the plasma membranes of target cells into the correct cellular compartment to exert a functional response. For example, exosomes derived from DCs can modulate the immune cell response by transferring peptide-loaded MHC class I and II cells complexed to DCs [78]. Another highly attractive feature as a drug delivery vehicle is the ability to home to target tissues. For example, melanoma exosomes home to sentinel nodes, demonstrating that exosomes do have intrinsic homing capability [79]. Exosomes loaded with anti-cancer drugs have already shown promise as a new therapeutic approach in animal models. The released exosomes loaded with cargo affect the target cells through the following mechanisms [80]. First, they activate certain signaling pathways of the target cells by interacting with specific ligand receptors. Next, the exosomes transfer surface receptors from one cell to another target cell by budding, followed by fusion with the plasma membrane. Then, they enter the cells using endocytic mechanisms such as receptor-mediated endocytosis, phagocytosis, and micropinocytosis and release their content into the cytoplasm. However, to use exosomes as biomarkers and DDSs, their isolation, purification, and characterization are extremely important and can be improved by using novel technologies.

### 5.2. Exosomes in the Tumor Microenvironment (TME)

The TME plays an important role in the proliferation and metastasis of tumor cells [81]. The TME comprises fibroblasts, stromal cells, and the ECM. Cancer-associated fibroblasts (CAF) and tumor-associated macrophages (TAM) are major cell populations in the stroma of all solid tumors and often exert protumorigenic functions [82,83]. Because CAF and TAM are known to modulate disease progression, we can expect that targeting cytokine and chemokine (e.g., CXCL, IL-6, and TGF-β) secretion by CAF could improve anti-cancer efficiency [84]. Several IL-6 inhibitors are already approved for immune disorders and are being investigated for their role in anti-cancer therapy. Exosomes can promote the formation of TME and also help in cell-to-cell communication in the TME by delivering proteins, nucleic acids, lipids, and signaling molecules (Figure 3). Moreover, exosomes are critical for tumor development due to their ten-fold higher secretory efficiency in cancer cells than in normal cells [85]. Thus, exosomes can release mRNAs and oncogenic proteins into target cells, which can fuse with the membrane and regulate tumor cell proliferation, invasion, and metastasis. Furthermore, exosomes from tumor cells induce adaptive changes in distant organs to create a “pre-metastatic” environment that is conducive to their growth and the formation of secondary metastatic foci [86].

Costa-Silva et al. found that exosomes derived from pancreatic cancer cells induce transforming growth factor β signaling, leading to the activation of hepatic stellate cells and ECM remodeling. In turn, fibronectin accumulation promotes an influx of bone marrow-derived macrophages (and potentially neutrophils) to the liver, providing a favorable niche for liver metastasis [87]. Breast cancer cell-derived exosomes play an important role in promoting breast cancer bone metastasis, which is associated with the formation of a pre-metastatic niche via transferring miR-21 to osteoclasts [88]. Because exosomes closely interact with the TME, by attaching CAF-targeting molecules or receptors, they can effectively reach cancer cells. Targeting CAFs or TAMs with exosomes could be of high impact for improving future targeted treatment strategies [89]. By contrast, HSPs mainly function as molecular chaperones. However, in cancer, they can suppress apoptosis, evade immune responses, and enhance angiogenesis and metastasis. Moreover, HSP also plays a role as a mediator of the resistance-associated secretory phenotype [90]. Hence, if possible, HSPs need not be incorporated in the production of exosomes to minimize such protumorigenic effects [91].

### 5.3. Engineering of Exosomes for Drug Delivery

Exosomes used as drug delivery vehicles have multiple advantages over existing synthetic systems. They have phospholipid bilayers, which can directly fuse with the plasma membrane of the target cell, thus improving the cellular internalization of the encapsulated drug. Targeted delivery of compounds to tumor vessels and tumor cells can enhance tumor detection and therapy. Docking-based (synaphic) targeting strategies use peptides, antibodies, and other molecules that bind to tumor vessels and tumor cells to deliver more drugs to tumors than to normal tissues [92]. A strategy to deliver drug-loaded exosomes to the tumor parenchyma is to use tumor-homing peptides such as iRGD, a novel cyclic peptide composed of 9-amino acids comprising an Arg-Gly-Asp (RGD) motif, on the surface. iRGD has a high binding affinity to αvβ3 and αvβ5 integrins abundant in tumor vasculatures [93]. Tian et al. found that combining DC-derived exosomes with specific iRGD peptides endows the exosomes with the ability to target breast cancer more efficiently than the chemical drug used alone [94]. Conversely, certain proteins or biomolecules with high affinity to normal cells (such as immune cells or other organ-specific cells) should be avoided during EV formation. One of the main issues with EV-based DDS is rapid clearance by mononuclear phagocyte systems. The most widely used “don’t eat me” signal is to bind PEG on the vesicle surface [95]. A recent report showed that surface modification using CD47 reduced uptake by RES [96]. In the same report, a cationized mannan-modified EV derived from DC2.4 cells was administered to saturate the MPS (eat me strategy) [95]. Alternatively, metalloproteinases that are naturally found in exosomes are another important component [97]. They can regulate the proteolytic activity in exosomes, thereby altering their contents. Moreover, they can degrade the ECM, which can enhance the efficiency of exosome-mediated drug delivery.

### 5.4. Clinical Applications of Artificial Exosomes

The role of exosomes in cancer initiation and progression is becoming increasingly apparent from preclinical and clinical investigations (summarized in the Table 3), and therefore, they are in the spotlight for potential use as cancer therapeutics [98]. With these characteristics, there are in vitro and clinical studies which show that anti-cancer drugs can be delivered more effectively when the drug is loaded into the exosome than when only the drug is administered.

Like other drugs, exosomes can be administered through various routes [99]. For in vivo analysis of exosome distribution, intravenous (IV) injection of exosomes was the dominant (78%) administration route, followed by intraperitoneal injection. The administration of exosomes through intranasal, hock, subcutaneous, and retro-orbital venous sinus routes was rare. The tissues with the most frequent accumulation of exosomes after IV injection were the liver, lung, spleen, and kidney.

## 6. Summary and Future Perspective

Exosomes as drug delivery vehicles possess huge advantages with low immunogenicity, long-term safety, and lack of cytotoxicity [62,100]. Conventional methods of delivering miRNAs, proteins, and chemical drugs show some limitations. For example, miRNAs are easily degraded in vivo, and chemical drugs are highly toxic to healthy cells. These obstacles can be solved by using exosomes as drug carriers. Currently, natural exosomes are used in preliminary clinical trials. Their translation, massive production, stabilized preparation, storage protocols, and quality control are challenges that must be overcome. As mentioned in a previous report, EV-based drug delivery remains challenging due to a lack of standardized isolation and purification methods, limited drug loading efficiency, and insufficient clinical-grade production [101]. Further development of cell-derived artificial exosomes and their engineering for isolation, purification, and drug loading will overcome these shortcomings. Artificial exosomes have commercial advantages for their up-scale productivity. Furthermore, by anchoring specific surface molecules on exosomes, we can increase the local concentration of exosomes at target cells or target disease sites, thereby reducing the toxicity and undesirable effects and maximizing therapeutic effects. The combination of artificial exosomes with anti-cancer drugs can lead to pivotal development in the treatment of cancer. In the future, novel and multifunctional artificial exosomes will be developed to improve healthcare. Therefore, further studies are needed to explore novel strategies of exosome-mediated therapies, particularly for cancer.

## Figures and Tables

**Figure 1 cells-11-00316-f001:**
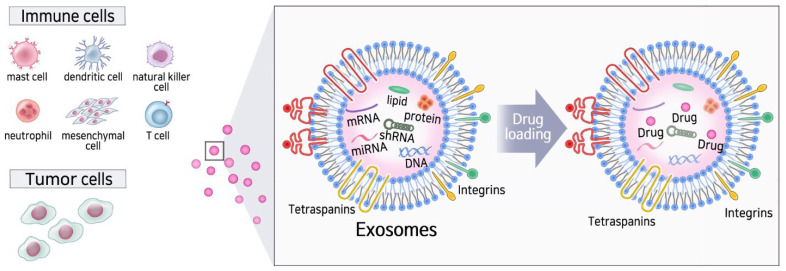
Diverse sources of exosomes and the effect on the immune system.

**Figure 2 cells-11-00316-f002:**
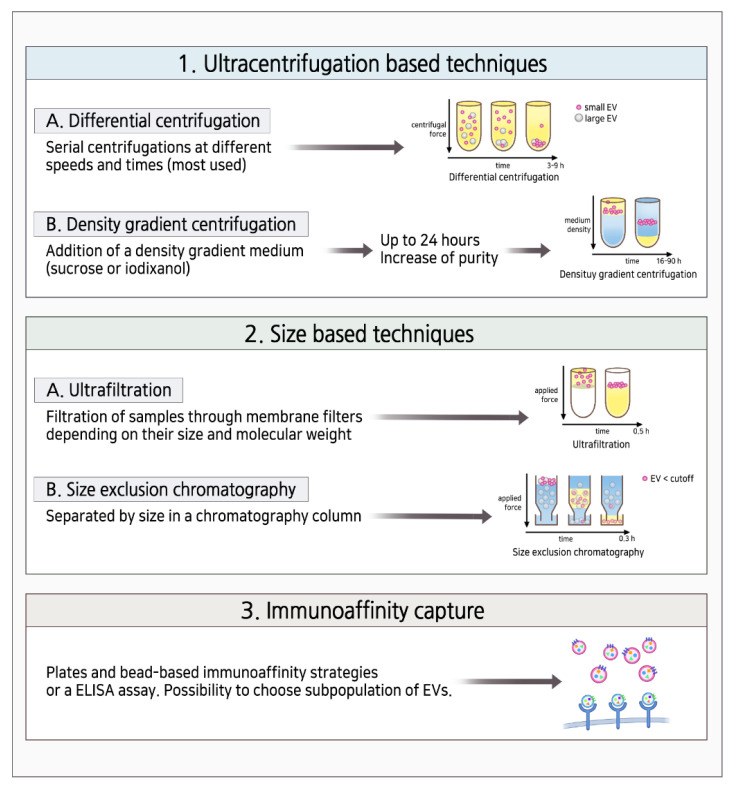
Schematic representation of the various methods used for exosome isolation.

**Figure 3 cells-11-00316-f003:**
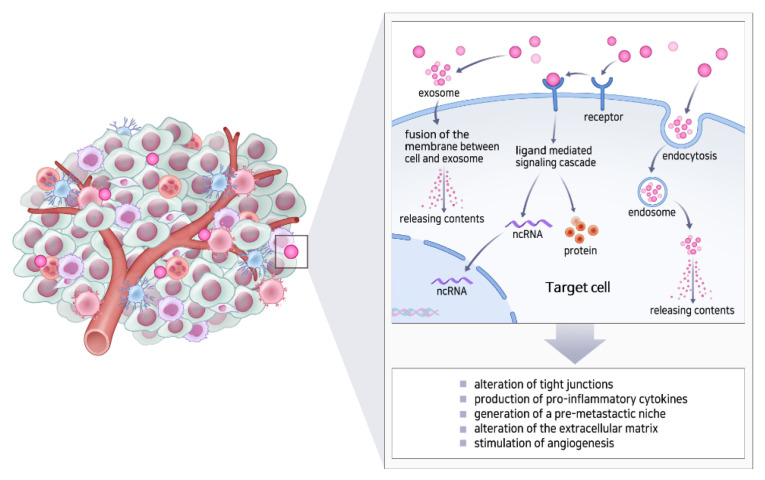
Strategies of exosomes to promote the formation of oncogenic microenvironment.

**Table 1 cells-11-00316-t001:** Pros and cons of the three methods for exosome isolation [74].

Method	Pros	Cons
Ultracentrifugation	Low cost, high purity, massive production	Time-consuming, mechanical damage, specialized equipment requirement
Ultrafiltration	Low cost, less time consuming, good portability	Moderate purity, mechanical damage, high cost
Immunological separation	High purity, no chemical contamination, simple	Small volume production, high cost

**Table 2 cells-11-00316-t002:** Various drug loading methods on exosomes and their efficiency [77].

Loading Method	Extracellular Vesicle (EV) Source	Loading Content	Loading Measurement	Efficiency (Type, %)	
Sonication	Raw 264.7 macrophages (mouse)	Paclitaxel (PTX)	High-performance liquid chromatography (HPLC)	Loading capacity	28.29 (SEM ± 1.38%)
	Raw 264.7 macrophages (mouse)	Dox	Fluorescence of Dox	Encapsulation efficiency	8.0–11.0%
	Raw 264.7 macrophages (mouse)	Catalase	Catalase enzymatic activity	Loading capacity	26.1 (SEM ± 1.2%)
Saponin permeabilization	Raw 264.7 macrophages (mouse)	Catalase	Catalase enzymatic activity	Loading capacity	18.5 (SEM ± 1.3%)
Mixing	Raw 264.7 macrophages (mouse)	Paclitaxel (PTX)	High-performance liquid chromatography (HPLC)	Loading capacity	1.4 (SEM ± 0.38%)
	LNCaP and PC-3 (human)	PTX	Ultra-performance liquid chromatography (UPLC)	Encapsulation efficiency	9.2% (SD ± 4.5%)
	Milk (bovine)	PTX	UPLC	Encapsulation efficiency	7.9 ± 1.0%
	Raw 264.7 macrophages (mouse)	Catalase	Catalase enzymatic activity	Loading capacity	4.9 (SEM ± 0.5%)
Electroporation	Raw 264.7 macrophages (mouse)	Paclitaxel (PTX)	High-performance liquid chromatography (HPLC)	Loading capacity	5.3 (SEM ± 0.48%)
	Immature dendritic cells (mouse)	Doxorubicin (Dox)	Fluorescence of Dox	Encapsulation efficiency	<20%
	Primary immature dendritic cells (mouse)	Glyceraldehyde 3-phosphate dehydrogenase	qPCR analysis, fluorescence microscopy	Encapsulation efficiency	10–38%
	Primary dendritic cells (mouse)	Vascular endothelial growth factor (VEGF) siRNA	qPCR analysis	Encapsulation efficiency	3%

**Table 3 cells-11-00316-t003:** Studies that investigated the use of exosomes for cancer therapy [77].

Source of Exosomes	Disease Type	Drugs	Isolation Methods
Raw 264.7 macrophages (mouse)	Multi-drug resistant cancers (in vitro and mouse models)	Doxorubicin and paclitaxel	Low-speed centrifugation with precipitating reagents and purifying column
Primary dendritic cells (mouse)	Breast cancer (in vitro and mouse models)	VEGF siRNA	Differential centrifugation and UC
Neutrophils	Malignant glioma	Doxorubicin	Ultracentrifugation
MSC	Colorectal cancer	Doxorubicin	Ultracentrifugation
Milk (bovine)	Lung cancer (in vitro and mouse models)	Paclitaxel	Differential gradient centrifugation and UC
MCF-7 breast carcinoma cells (human)	Breast carcinoma (in vitro)	Doxorubicin	Differential gradient centrifugation
LNCaP and PC-3 prostate cancer cells (human)	Prostate cancer (in vitro)	Paclitaxel	Differential centrifugation
Lewis lung carcinoma cells (mouse)	Lung cancer (in vitro)	Methotrexate	Differential gradient centrifugation
Immature dendritic cells (mouse)	Breast cancer (in vitro and mouse models)	Doxorubicin	Ultrafiltration, UC, and gradient centrifugation
HeLa cervical cancer cells (human)	Cervical cancer (in vitro)	Dextran	Precipitating reagents (total exosome isolation kit, Invitrogen)
H22 hepatocarcinoma cells (mouse)	Hepatocarcinoma (in vitro and mouse models)	Cisplatin	Differential gradient centrifugation
Gastric cancer (SKBR-3)	Gastric cancer	Trastuzumab	Ultracentrifugation
EL-4 lymphoma cells (mouse)	Tumor-induced inflammation (in vitro and mouse models)	Curcumin	Sucrose gradient centrifugation
Bone-marrow-derived MSCs (human)	Lung cancer (in vitro)	TRAIL	Filtration
	Pleural mesothelioma (in vitro)	TRAIL	Filtration
	Renal cancer (in vitro)	TRAIL	Filtration
	Breast adenocarcinoma (in vitro)	TRAIL	Filtration
	Neuroblastoma (in vitro)	TRAIL	Filtration
B16-F10 melanoma cells (mouse)	Melanoma (in vitro)	Superparamagnetic iron oxide nanoparticles	Ultracentrifugation (UC)
B16BL6 melanoma cells (mouse)	Melanoma (in vitro and mouse models)	CpG DNA	Filtration and differential UC
ADR/MCF-7 breast carcinoma cells (human)	Breast carcinoma (in vitro)	Cisplatin	Differential gradient centrifugation
A549 lung carcinoma cells (human)	Lung carcinoma (in vitro, mouse models, and stage IV human patients)	Doxorubicin	Differential gradient centrifugation
M1 macrophage	Pancreatic cancer	Gemcitabine/Deferasirox	Ultracentrifugation
Human breast cancer cell line (EFM-192A)	Breast cancer	Trastuzumab	Ultracentrifugation

## Data Availability

Not applicable.

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
