# Peer review of "Exosome as a Delivery Vehicle for Cancer Therapy"

_cells, 2022, doi:10.3390/cells11030316_

Round 1

Reviewer 1 Report

Exosomes / sEV (small EVs) as a delivery vehicle for cancer therapy

Major questions

  1. What is the best resource for sEV? Cancer cells, immune cells, MSC, milk, acerola, bacterial OMV, or others? Why? Which is appropriate or inappropriate?
  2. How can the drug-loaded EVs be administrated into living bodies, such as oral, intravenous (iv), etc.?
  3. What factors should or should not be loaded in/on the surface of EVs for specific targeting on cancer cells?
  4. Can sEV target immune checkpoints or tumor microenvironmental cells such as CAF, TAM, or tumor angiogenesis? How?
  5. How are "eat me" signal and "don’t eat me" signal involved in sEV-based DDS. How can we control and overcome these signals when using EVs as DDS?
  6. What are the potential side effects and adverse effects when using sEV as DDS? How can we overcome those limitations?
  7. What are the stabilizing and destabilizing factors for sEV in vivo and in vitro?
  8. How can metalloproteinases and their inhibitors alter the function of sEV as DDS?
  9. How can heat shock proteins (HSP) be involved in the functions of sEV as DDS?
  10. How can we isolate homogenous EVs for DDS applications from heterogenous EVs?
  11. What is the best method to load drugs into EVs? The methods include sonication, electroporation, transfection reagent, saponin, mixing, freeze/thaw, extrusion (Theranostics 2019 Review). List up these in a table and compare features such as pros and cons.
  12. Can sEV/exosome-based DDS overcome therapy resistance, including EV-based drug resistance? Ref: Biology 2020, 9, 47
  13. Fig 1
    1. The scheme and text are too small and complicated. Enlarge them.
    2. How do the loaded drugs switch the immunosuppressive EVs to the immunostimulatory EVs? Which drug can induce immunostimulatory effects? Do some drugs play an anti-tumor / anti-cancer effect?
    3. Other cancer-related cell types could be added, e.g., T cells.
  14. Fig 2: Various methods of exosome isolation were illustrated.
    1. How different are these methods? List up their features, pros, cons.
    2. How should sEV or exosomes be isolated when used as DDS? Why?
  15. Table 1: Studies using sEV for cancer therapy were listed up.
    1. More studies were reported in Theranostics 2019 Review 9, 26, Cancers 2021 Review 13, 326.
    2. Is there any paper reporting miRNA, ncRNA, CRISPR/gRNA (Nat Commun 2020, 11,1334), and others as nucleic acid drugs loaded in sEV?
    3. Cancer cell-derived EVs / exosomes could contain oncogenic factors. Could these EVs/exosomes be useful as DDS? If not, are immune cells, MSC, fruits, milk, and bacteria more hopeful as resources for EVs?
  16. In section 3.2, differences between "exosomes and nanoparticles in the penetration into solid tumor tissues" are described. Cite some papers showing exosomes / EV penetration into tumors in vitro and in vivo.
  17. In section 4, three methods were described twice, but size exclusion chromatography (SEC) is lacking. Add the SEC. Is the SEC useful for the separation of heterogenous EVs and the purification of exosomes? Cite some original articles.

Minor points

  1. In the abstract
    1. "Exosome is a type of micro vesicle that…", but exosomes are not microvesicles. The sentence is confusing and could be rephrased.
    2. The phrase "load any kinds of molecules" is incorrect and could be rephrased.
  2. Section 1.2, on the bottom, "inter and intracellular communication": What is the intracellular communication" meaning?
  3. At the bottom of page 3, "HSP70-90" could be HSP70 and HSP90.
  4. Section 5.1, line 4: "Nature and help cell…" is a wired sentence and could be corrected.

Author Response

Point-by point responses to Reviewer #1

Major questions

1. What is the best resource for sEV? Cancer cells, immune cells, MSC, milk, acerola, bacterial OMV, or others? Why? Which is appropriate or inappropriate?

- Because EVs has characteristics of its cell of origin, the best resource of sEVs depends on its target cells/tissues. Among them, MSC derived exosomes are frequently used nowadays as it is less immunogenic. MSC derived exosomes do not contain class I and class II human major histocompatibility complex(MHC) proteins or co-stimulatory molecules such as CD80 or CD86, which makes it to evade immune systems. We added it in the section 1.2.

2. How can the drug-loaded EVs be administrated into living bodies, such as oral, intravenous (iv), etc.?

- Like other drugs, exosomes can be administered through various route[95]. For in vivo analysis of exosome distribution, intravenous (IV) injection is a dominant (78%) administration route, followed by intraperitoneal injection. The administration of exosomes through intranasal, hock, subcutaneous, and retro-orbital venous sinus routes are rarely used. The major tissues that accumulate exosomes was reported as the liver, lung, spleen, and kidney. We added this in the section 5.4.

3. What factors should or should not be loaded in/on the surface of EVs for specific targeting on cancer cells?

- Targeted delivery of compounds to tumor vessels and tumor cells can enhance tumor detection and therapy. Docking-based (synaptic) targeting strategies use peptides, antibodies, and other molecules that bind to tumor vessels or tumor cells to deliver more drug to tumors than normal tissues. One of a strategy to deliver drug loaded exosomes to tumor parenchyma is to use tumor homing peptide such as iRGD, a novel cyclic peptide composed of 9-amino acids including an Arg-Gly-Asp (RGD) motif, on the surface . iRGD has a high binding affinity to αvβ3 and αvβ5 integrins abundant in tumor vasculatures. Y. Tian et al. found that combining DCs-derived exosomes with specific IRGD peptides enhanced the ability of exosome for targeting breast cancer more efficiently. Conversely, certain protein or biomolecules with high affinity to normal cells (such as immune cells or other organ specific cells) should be avoided in EV formation. We added this in the section 5.3.

4. Can sEV target immune checkpoints or tumor microenvironmental cells such as CAF, TAM, or tumor angiogenesis? How?

- There have been reports that highlights the exosome mediated-crosstalk between cancer cell and CAF or cancer cell and TAM. However, to our knowledge, there is no known exosome that specifically targets CAF or TAM. By attaching CAF / TAM targeting molecules or receptors on exosomes, we think it can reach these cells effectively. We added this in the section 5.2.

5. How are "eat me" signal and "don’t eat me" signal involved in sEV-based DDS. How can we control and overcome these signals when using EVs as DDS?

- One of the main issues with sEV-based DDS is rapid clearance by mononuclear phagocyte systems. The most widely used “don’t eat me” signal is to bind PEG on the surface of vesicles. A recent report showed the surface modification using CD47 has proven to decrease uptake by Reticuloendothelial systems (RES). In the same report, a cataionized mannan-modified extracellular vesicles derived from DC2.4 cells were administered to saturate the MPS (eat me strategy). We added this in the section 5.3

6. What are the potential side effects and adverse effects when using sEV as DDS? How can we overcome those limitations?

- As nicely mentioned in a previous report, the EV-based drug delivery remains challenging, due to lack of standardized isolation and purification methods, limited drug loading efficiency, and insufficient clinical grade production. Further development of cell derived artificial exosome and its engineering for isolation, purification and drug loading will overcome these. We added this in the section 6.

7. What are the stabilizing and destabilizing factors for sEV in vivo and in vitro?
- Their stability depends on contents of buffer, temperature, pH in vivo. Therefore, we should
 care the storage, modification conditions.

8. How can metalloproteinases and their inhibitors alter the function of sEV as DDS?

- Metalloproteinases are naturally found in exosomes. It can regulate proteolytic activity in exosome, thereby alter its contents. Moreover, as a drug delivery vehicle, it can degrade extracellular matrix (ECM) that can enhance the efficiency of exosome mediated drug delivery. We added this in the section 5.3

9. How can heat shock proteins (HSP) be involved in the functions of sEV as DDS?

- Heat shock proteins (HSP) are mainly function as a molecular chaperone. However, in cancer it also can suppresses apoptosis, evade immune response and enhance angiogenesis and metastasis. Moreover, HSP also plays as a mediator of Resistance-Associated Secretory Phenotype (RASP). Hence, if possible, the HSP need to be in the production of exosome to minimize such protumorigenic effect. We added this point in the section 5.2

10. How can we isolate homogenous EVs for DDS applications from heterogenous EVs?

- Among various isolation methods for EV, we think size exclusion chromatography (SEC) is most effective way to obtain homogenous EV by far. The SEC is also reported to remove soluble protein contaminants, and relatively easy to scale-up for manufacturing clinical grade product. We added this in the section 4.1

11. What is the best method to load drugs into EVs? The methods include sonication, electroporation, transfection reagent, saponin, mixing, freeze/thaw, extrusion (Theranostics 2019 Review). List up these in a table and compare features such as pros and cons.
- From the excellent review that reviewer provided, we revised our table so that the loading efficiency can be added (Table 1). Also, we added description as follows. Table 1 summarizes various drug loading methods on exosome. Considering previous results of efficiency measured, sonication seems to work well in macrophage derived exosomes whereas electroporation seems better for primary dendritic cell-derived exosome.

12. Can sEV/exosome-based DDS overcome therapy resistance, including EV-based drug resistance? Ref: Biology 2020, 9, 47

  • From the important review that reviewer provided, we learned that EV could induce resistance-associated secretory phenotype (RASP). We agree that this can be a novel mechanism of drug resistance. As suggested in the review, we think the strategy to overcome this would be suppress EMT or stem cell like conversion of the cancer cell. Also, MSC or macrophage-derived exosome can be tried as well.

13. Fig 1

1. The scheme and text are too small and complicated. Enlarge them.

    >> We maximized the scheme and enlarged text

2. How do the loaded drugs switch the immunosuppressive EVs to the immunostimulatory EVs? Which drug can induce immunostimulatory effects? Do some drugs play an anti-tumor / anti-cancer effect?
: This figure seems to be confusing.

>> We regret that the figure was confusing. Now, we simplified the figure to clearly highlights the role of EV as a drug carrier.

3. Other cancer-related cell types could be added, e.g., T cells.

>> We added T cells in the figure 1.

14. Fig 2: Various methods of exosome isolation were illustrated.

    1. How different are these methods? List up their features, pros, cons.

>> Following the comment, we added a Table1 in the revision, as shown below.

Method

Pros

Cons

Ultracentrifugation

Low  cost, high purity, massive production

Time consuming, mechanical damage, high equipment requirement

Ultrafiltration

Low cost, less time consuming,

Good portability

Moderate purity, mechanical damage, high cost

Immunological separation

High purity, no chemical contamination, Simple

Small volume production,

High cost

2. How should sEV or exosomes be isolated when used as DDS? Why?

- A recent report nicely presented that for the clinical trials of exosome, the mostly applied method is tangential flow fractionation (TFF) combined with ultracentrifugation. These methods are used as it maximized purity, uniformity and integrity of the exosome. We added this in the section 4.1

15. Table 3: Studies using sEV for cancer therapy were listed up.

    1. More studies were reported in Theranostics 2019 Review 9, 26, Cancers 2021 Review 13, 326. >> We appreciate the helpful references and now added more studies in the Table 3.
    2. Is there any paper reporting miRNA, ncRNA, CRISPR/gRNA (Nat Commun 2020, 11,1334), and others as nucleic acid drugs loaded in sEV? >> As the reviewer pointed out, a recent study reported the incorporation of CRISPR/gRNA into exosome. In addition to that, siRNA, aptamer and ASO (antisense oligonucleotide. This is added in the section 3 with citations.
    3. Cancer cell-derived EVs / exosomes could contain oncogenic factors. Could these EVs/exosomes be useful as DDS? >> We don’t think these oncogenic factors would be useful as a DDS, because these factors can transform normal cell unless the exosome is specific to the cancer (or other target) cells.
    4. If not, are immune cells, MSC, fruits, milk, and bacteria more hopeful as resources for EVs? >> This is an interesting question, as we have not reviewed milk or bacteria as an exosome source. Certainly, normal cell-driven exosome would be safer for DDS. However, milk or bacteria can cause undesirable immune response, so that it should be carefully addressed.

16. In section 3.2, differences between "exosomes and nanoparticles in the penetration into solid tumor tissues" are described. Cite some papers showing exosomes / EV penetration into tumors in vitro and in vivo. >> We added a recent article showing a biocompatible tumor-cell-derived exosome-biomimetic porous silicon nanoparticles (PSiNPs) that penetrating tumor tissue efficiently (Nature Communications volume 10, Article number: 3838 (2019)).

17. In section 4, three methods were described twice, but size exclusion chromatography (SEC) is lacking. Add the SEC. Is the SEC useful for the separation of heterogenous EVs and the purification of exosomes? Cite some original articles. >> We agree that the SEC is a useful method for the isolation of exosome, so that we added it in the section 4.1 (now combined with other methods without subheadings). Also, we cited an original article for this technique (Clin Cancer Res, 2005 Feb 1;11(3):1010-20), where Fas ligand-positive microvesicles were isolated from sera of patients.

Minor points

18. In the abstract

    1. "Exosome is a type of micro vesicle that…", but exosomes are not microvesicles. The sentence is confusing and could be rephrased. >> We regret the sentence was confusing. Now we revised it as “Exosome is a small vesicle ~”
    2. The phrase "load any kinds of molecules" is incorrect and could be rephrased. >> We agree that the phrase was wrong. Now we revised it as “load many kinds of molecules ~”

19. Section 1.2, on the bottom, "inter and intracellular communication": What is the intracellular communication" meaning? >> We recognize this is incorrect. Now we revised it as “intercellular communication”

20. At the bottom of page 3, "HSP70-90" could be HSP70 and HSP90. >> That is correct. We revised it as HSP70/HSP90.

21. Section 5.1, line 4: "Nature and help cell…" is a wired sentence and could be corrected.

>> We regret the phrase was wrong. It is thus deleted.

Reviewer 2 Report

The Manuscript by Authors Moon and Chang; is a mini-review, relevant to the field topic to review. The study describes the exosome as drug delivery vehicles for cancer therapy and their basic properties. Then, Cell type-specific exosomes, Artificial exosome, and their separation methods. Finally, about Exosomes in the tumour microenvironment and. Clinical application of artificial exosome. The study requires major revision before it is accepted in order to make the study more impactful.

Under cell-type-specific exosomes, subheadings too little were discussed and relevant discussion to the title is missing.

No connections between subheadings are missing. The title is exosome as a delivery vehicle for cancer therapy, but the content is half of it, not relevant. As author mainly focuses on artificial exosomes second part of the manuscript. Then describing the isolation of artificial exosomes.

Drug loading was not detailed discussed.

The review looks like they touched all topics with a pinch of salt.

This review required a complete revision of the manuscript with relevant and well-connected subheadings. 

Author Response

Point by point responses to Reviewer #2

The Manuscript by Authors Moon and Chang; is a mini-review, relevant to the field topic to review. The study describes the exosome as drug delivery vehicles for cancer therapy and their basic properties. Then, Cell type-specific exosomes, Artificial exosome, and their separation methods. Finally, about Exosomes in the tumour microenvironment and. Clinical application of artificial exosome. The study requires major revision before it is accepted in order to make the study more impactful.

>> We sincerely appreciate for all valuable comments and critical points raised by the reviewer #2

Under cell-type-specific exosomes, subheadings too little were discussed and relevant discussion to the title is missing.

>> We agree to the reviewer’s point. As we intended to focus on the therapeutic potential of exosome (as a drug delivery vehicle), we combined these subheadings in the section 2 into one section. Instead, we added more contents in the section 5. Please understand this change and we appreciate for the helpful comment.

No connections between subheadings are missing. The title is exosome as a delivery vehicle for cancer therapy, but the content is half of it, not relevant. As author mainly focuses on artificial exosomes second part of the manuscript. Then describing the isolation of artificial exosomes.

>> We regret that our manuscript lacked connections among subheadings and did not match with the title. Following the reviewers’ comment, we revised subheadings, combined short subheadings and added section 5 to describe the current information and ongoing studies on exosome as a drug delivery vehicle.

Drug loading was not detailed discussed.

>> We appreciate for the critical comment. To address this point, we revised our Table 2 in the section 4.2, that now covers loading methods and its cellular source, loading content and loading efficiency. According to this information, we discussed sonication and electroporation as suitable methods for the macrophage and dendritic cells, respectively.

The review looks like they touched all topics with a pinch of salt.

>> We respectfully state that the aim of this review is to cover the overall topics of exosomes, to provide background information of the exosome to the diverse scope of readership.

This review required a complete revision of the manuscript with relevant and well-connected subheadings. 

>> We thank for the helpful comment. Following the comment, we fully revised the subheadings as follows.

  1. Introduction
  2. Natural, cell-type specific exosomes
  3. Artificial Exosome as a drug delivery vehicle
  4. Purification and drug loading on exosomes
  5. Therapeutic Aspects of Exosome as a Drug Delivery System (DDS)
  6. Summary and future perspective

Round 2

Reviewer 1 Report

Reviewer #1 Report Round 2

The authors answered many points appropriately. However, many points are still deficient.

Major points

  1. What is the best resource for sEV? Cancer cells, immune cells, MSC, milk, acerola, bacterial OMV, or others? Why? Which is appropriate or inappropriate?

- Because EVs has characteristics of its cell of origin, the best resource of sEVs depends on its target cells/tissues. Among them, MSC derived exosomes are frequently used nowadays as it is less immunogenic. MSC derived exosomes do not contain class I and class II human major histocompatibility complex(MHC) proteins or co-stimulatory molecules such as CD80 or CD86, which makes it to evade immune systems. We added it in the section 1.2.

I understand the advantage of the MSC-sEV for DDS application, as "MSC-exosomes do not contain class I and class II human MHC proteins or co-stimulatory molecules such as CD80 or CD86, which makes it to evade immune systems. I strongly recommend to make a figure that emphasize this molecular basis.

At the same time, CD80 and CD86 are found on the surface of immune cells such as M1 macropahges (Mph) and DCs, which can produce CD80/CD86-sEV, but Mph-sEV and DC-sEV are also used as DDS, as authors showed in tables. Is the expression of CD80 and 86 on the sEV meaningful for cancer therapy or a problem? Make a figure comparing MSC-sEV vs immune cell-sEV.

Also, in Fig 1, the authors categorized the mesenchymal cells in the immune cells, but they are not immune cells. And, the authors have emphasized MSC is the best source for exosomes for DDS, but MSC is not found in Fig 1. Re-classify the cell types, including MSC in Fig 1.

  1. What factors should or should not be loaded in/on the surface of EVs for specific targeting on cancer cells?
    5. How are "eat me" signal and "don’t eat me" signal involved in sEV-based DDS. How can we control and overcome these signals when using EVs as DDS?

- The authors answered about the docking-based targeting strategies, iRGD binding to specific integrins and DC-sEV. Also the authors answered about the PEG, CD47, and DC2.4-sEV as don’t eat me signal strategy in section 5.3. I strongly recommend to make a figure showing these systems.

  1. How can metalloproteinases and their inhibitors alter the function of sEV as DDS?

- Metalloproteinases are naturally found in exosomes. It can regulate proteolytic activity in exosome, thereby alter its contents. Moreover, as a drug delivery vehicle, it can degrade extracellular matrix (ECM) that can enhance the efficiency of exosome mediated drug delivery. We added this in the section 5.3.

Altough the authors answered in the letter above, I could not find the sentences and references in the Ms. Add these appropriately.

  1. How can heat shock proteins (HSP) be involved in the functions of sEV as DDS?

- Heat shock proteins (HSP) are mainly function as a molecular chaperone. However, in cancer it also can suppresses apoptosis, evade immune response and enhance angiogenesis and metastasis. Moreover, HSP also plays as a mediator of Resistance-Associated Secretory Phenotype (RASP). Hence, if possible, the HSP need to be in the production of exosome to minimize such protumorigenic effect. We added this point in the section 5.2

Add references in the sentence " However, in cancer it also can suppresses apoptosis, evade immune response and enhance angiogenesis and metastasis."

  1. Fig 2: Various methods of exosome isolation were illustrated.
    1. How different are these methods? List up their features, pros, cons.

>> Following the comment, we added a Table1 in the revision, as shown below.

According to the comments, authors added Table 1. However, the table is too simple and important methods are lacked. Add density gradient centrifugation, SEC, and TFF in the table.

Also, enlarge Fig 2.

  1. How should sEV or exosomes be isolated when used as DDS? Why?

- A recent report nicely presented that for the clinical trials of exosome, the mostly applied method is tangential flow fractionation (TFF) combined with ultracentrifugation. These methods are used as it maximized purity, uniformity and integrity of the exosome. We added this in the section 4.1

  1. Table 3: Studies using sEV for cancer therapy were listed up.
    1. More studies were reported in Theranostics 2019 Review 9, 26, Cancers 2021 Review 13, 326. >> We appreciate the helpful references and now added more studies in the Table 3.

According to the comment, Table 3 was updated largely, but this might be a copy-and-paste from the ref 77. Why did the authors remove many evidences listed in the first Ms, such as A549, 4T1/HER2 BCa cells, MSC544, Panc1, EFM-192? Also, I recommend clasify the source of exosomes in the table, e.g. MSC, immune cells, cancer cells..

Citing ref 77 in the titles of table 3 and 2 is not necessary.

Minor points

  1. In the abstract
    1. "Exosome is a type of micro vesicle that…", but exosomes are not microvesicles. The sentence is confusing and could be rephrased. >> We regret the sentence was confusing. Now we revised it as “Exosome is a small vesicle ~”

In the ISEV society, small extracellular vesicles (sEV) is recently recommended to use, instead of exosomes. Therefore, correct the sentence as “Exosome is a small extracellular vesicle (sEV) ~”

Additional points

  1. In Fig 3, on the right, how can the components in the endosome be released to cytoplasm? Is there known mechanism of "endosomal escape"? How can the endosomal molecules escape from lysosomal degradation, autophagy, and recycling endosome?

Reviewer 2 Report

The authors revised the manuscript according to comments and suggestions. 

Author Response

We are grateful for the comments of reviewer 2